# Signal peptide represses GluK1 surface and synaptic trafficking through binding to amino-terminal domain

Gui-Fang Duan[1], Yaxin Ye[2], Sha Xu[2], Wucheng Tao[3], Shiping Zhao[2], Tengchuan Jin [4], Roger A. Nicoll[3,5], Yun Stone Shi [1,6] & Nengyin Sheng [2,7]

Kainate-type glutamate receptors play critical roles in excitatory synaptic transmission and synaptic plasticity in the brain. GluK1 and GluK2 possess fundamentally different capabilities in surface trafficking as well as synaptic targeting in hippocampal CA1 neurons. Here we find that the excitatory postsynaptic currents (EPSCs) are significantly increased by the chimeric GluK1(SP$^{GluK2}$) receptor, in which the signal peptide of GluK1 is replaced with that of GluK2. Coexpression of GluK1 signal peptide completely suppresses the gain in trafficking ability of GluK1(SP$^{GluK2}$), indicating that the signal peptide represses receptor trafficking in a *trans* manner. Furthermore, we demonstrate that the signal peptide directly interacts with the amino-terminal domain (ATD) to inhibit the synaptic and surface expression of GluK1. Thus, we have uncovered a trafficking mechanism for kainate receptors and propose that the cleaved signal peptide behaves as a ligand of GluK1, through binding with the ATD, to repress forward trafficking of the receptor.

[1] State Key Laboratory of Pharmaceutical Biotechnology, Department of Neurology, Nanjing Drum Tower Hospital, The Affliated Hospital of Nanjing University Medical School, and Minister of Education Key Laboratory of Model Animal for Disease Study, Model Animal Research Center, Nanjing University, Nanjing 210032, China. [2] State Key Laboratory of Genetic Resources and Evolution, Kunming Institute of Zoology, Chinese Academy of Sciences, Kunming 650223, China. [3] Department of Cellular and Molecular Pharmacology, University of California, San Francisco 94143 CA, USA. [4] Hefei National Laboratory for Physical Sciences at Microscale, CAS Key Laboratory of Innate Immunity and Chronic Disease, School of Life Sciences and Medical Center, University of Science and Technology of China, Hefei 230007, China. [5] Department of Physiology, University of California, San Francisco 94143 CA, USA. [6] Institute for Brain Sciences, Nanjing University, Nanjing 210032, China. [7] Center for Excellence in Animal Evolution and Genetics, Chinese Academy of Sciences, Kunming 650223, China. Correspondence and requests for materials should be addressed to Y.S.S. (email: yunshi@nju.edu.cn) or to N.S. (email: shengnengyin@mail.kiz.ac.cn)

Glutamate is the principal excitatory neurotransmitter in the brain and mediates synaptic transmission through three distinct types of ionotropic glutamate receptors: AMPA, NMDA, and kainate receptors (KARs)[1]. In contrast to the widely expressed AMPA receptors and NMDA receptors at glutamatergic synapses, KARs are expressed specifically at a subset of synapses[2,3]. KARs are assembled from combinations of five subunits GluK1-5. The low-affinity GluK1-3 subunits are obligatory and able to form homomeric channels, while the high-affinity GluK4/5 can only form functional heteromeric receptors with GluK1-3[2,4]. Much of our knowledge about synaptic KARs is based on studying excitatory mossy fiber synapses onto CA3 pyramidal cells[5]. There these receptors are localized both post-synaptically and presynaptically, and are responsible for a slow excitatory postsynaptic current (EPSC)[6,7] and also are involved in the profound frequency facilitation of these synapses[8–11], respectively. Although functional KARs are expressed on the surface of hippocampal CA1 pyramidal neurons, the Schaffer collateral-CA1 synapses are devoid of KAR-mediated synaptic responses[6,12–14]. Therefore, these synapses provide a null background system to study the rules governing KAR synaptic function. We recently revealed that GluK1 and GluK2 receptors are fundamentally different in terms of their forward trafficking abilities. Both surface expression and synaptic trafficking of the GluK1 receptor require the auxiliary Neto proteins, while GluK2 itself traffics to the surface and the synapse independent of Neto proteins[14–16]. These findings raise questions as to what determines the specificity of KARs trafficking properties.

All KAR subunits share a common topology and previous studies focused on the role of their cytoplasmic C-terminal domains (CTDs) for receptor trafficking[2,17]. Recently, several studies uncovered an unexpected role of the extracellular amino-terminal domain (ATD) for GluK2 synaptic targeting[15,18,19], and we further discover that it is the amino-terminal regions (ATRs, including signal sequence and ATD) that control the different trafficking properties between GluK1 and GluK2[15]. However, the ATR sequences between GluK1 and GluK2 are quite conserved except for regions around N-terminal signal sequences. We thus extended our study of the ATRs to determine the minimal structural features responsible for the different trafficking capabilities between GluK1 and GluK2.

Signal sequences are N-terminal amino acid residues, ranging from 15 to more than 50, of newly synthesized secretory or membrane proteins[20]. In eukaryotes, signal sequences direct the insertion of nascent proteins into the membrane of the endoplasmic reticulum (ER) and are then usually cleaved off by signal peptidase, resulting in free signal peptides. Besides the well-characterized roles in ER targeting and membrane insertion[20,21], some signal peptides have post-targeting functions, either as transmembrane peptides, or released into the cytosol or ER lumen after intramembrane proteolysis[22]. Recently, we have found that the signal peptide of AMPA receptor subunit GluA1 has an unconventional function of regulating the subunit spatial position for heteromeric GluA1/A2 receptor assembly[23], suggesting that signal peptides of glutamate receptors might have other cellular and molecular functions in addition to their canonical ER targeting roles.

Using the null background system of excitatory synapses onto CA1 pyramidal cells, we find an inhibitory regulation of GluK1 trafficking by its signal peptide. In a *trans* manner, the cleaved signal peptide interacts with the ATD, thereby restraining the receptor's expression at both the neuronal surface and synapses. Our work thus demonstrates that the signal peptide of GluK1 has

a ligand-like effect and interacts directly with the ATD in the regulation of GluK1 trafficking.

## Results

**Signal peptide inhibits the synaptic targeting of GluK1.** We have previously revealed that GluK1 is unable to traffic to synapses by itself, while GluK2 does, and swapping the extracellular ATRs switches their distinct synaptic trafficking properties[15]. To further dissect the underlying molecular mechanisms, we made chimeric receptors by swapping the subregions of the ATRs between GluK1 and GluK2. We used organotypic rat hippocampal slices and exogenously expressed these mutated KARs into CA1 neurons through biolistic transfection. We then measured the synaptic responses of transfected and neighboring wild-type CA1 neurons by simultaneous dual whole-cell recordings. Compared with wild-type GluK1, expression of the chimeric GluK1(SP$^{GluK2}$) receptor, in which the GluK2 signal peptide (SP$^{GluK2}$) was used to replace that of GluK1 (Fig. 1a and Supplementary Fig. 1a), dramatically increased synaptic responses (Fig. 1a, e), indicating successful synaptic trafficking of the mutated GluK1 receptor. Moreover, the enhancement by GluK1(SP$^{GluK2}$) is similar in magnitude to that of previously reported GluK1(ATR$^{GluK2}$) (Fig. 1e)[15], demonstrating that the signal peptide is the critical region regulating GluK1 trafficking. This result is intriguing, as signal peptides of glutamate receptors are generally recognized as a location code but not to have other physiological functions. Because GluK2 harbors a strong synaptic targeting capability in CA1 neurons[15], it is reasonable to suspect that the GluK2 signal peptide endows GluK1 with an artificial trafficking ability. To test this possibility, we replaced the signal peptide of GluK1 with that of GluA1, a glutamate receptor having no significant effect on synaptic transmission when overexpressed in CA1 neurons[24]. The resultant GluK1(SP$^{GluA1}$) chimeric receptor also increased the EPSCs significantly (Fig. 1b, e). These data thus pointed to another possibility: the GluK1 receptor is capable of synaptic targeting but is inhibited by its signal peptide, replacement of which by that of GluK2 or GluA1 releases this inhibition.

If this assumption is correct, reintroduction of the GluK1 signal peptide should suppress the enhanced trafficking of GluK1(SP$^{GluK2}$). Therefore, we coexpressed the GluK1 signal peptide together with the GluK1(SP$^{GluK2}$) chimeric receptor in the same neurons and then examined the effect on synaptic transmission. To achieve this, we constructed SP$^{GluK1}$-GFP, which should direct the synthesized GFP to the secretary pathway and leave the GluK1 signal peptide in ER after cleavage by signal peptidase. The SP$^{GluK1}$-GFP was inserted before IRES-GluK1 (SP$^{GluK2}$) to ensure that SP$^{GluK1}$ expression level was in excess of GluK1(SP$^{GluK2}$) expression (Fig. 1d). We found a control construct GFP-IRES-GluK1(SP$^{GluK2}$) significantly increased synaptic responses in CA1 cells (Fig. 1c), while this potentiation was fully reversed by the coexpressed GluK1 signal peptide in SP$^{GluK1}$-GFP-IRES-GluK1(SP$^{GluK2}$) (Fig. 1d, f). Taken together, these results suggest that the signal peptide represses GluK1 receptor synaptic trafficking in a *trans* manner.

**GluK1 signal peptide is excised from the native receptor.** Conventional signal peptides are cleaved in ER and then degraded. However, uncleaved pseudo signal peptides have been reported to regulate GPCR receptors trafficking[25]. Since the trafficking of GluK1 and GluK2 are profoundly different and the GluK1 signal peptide specifically inhibits its trafficking, we next examined whether the cleavage of GluK1 and GluK2 signal

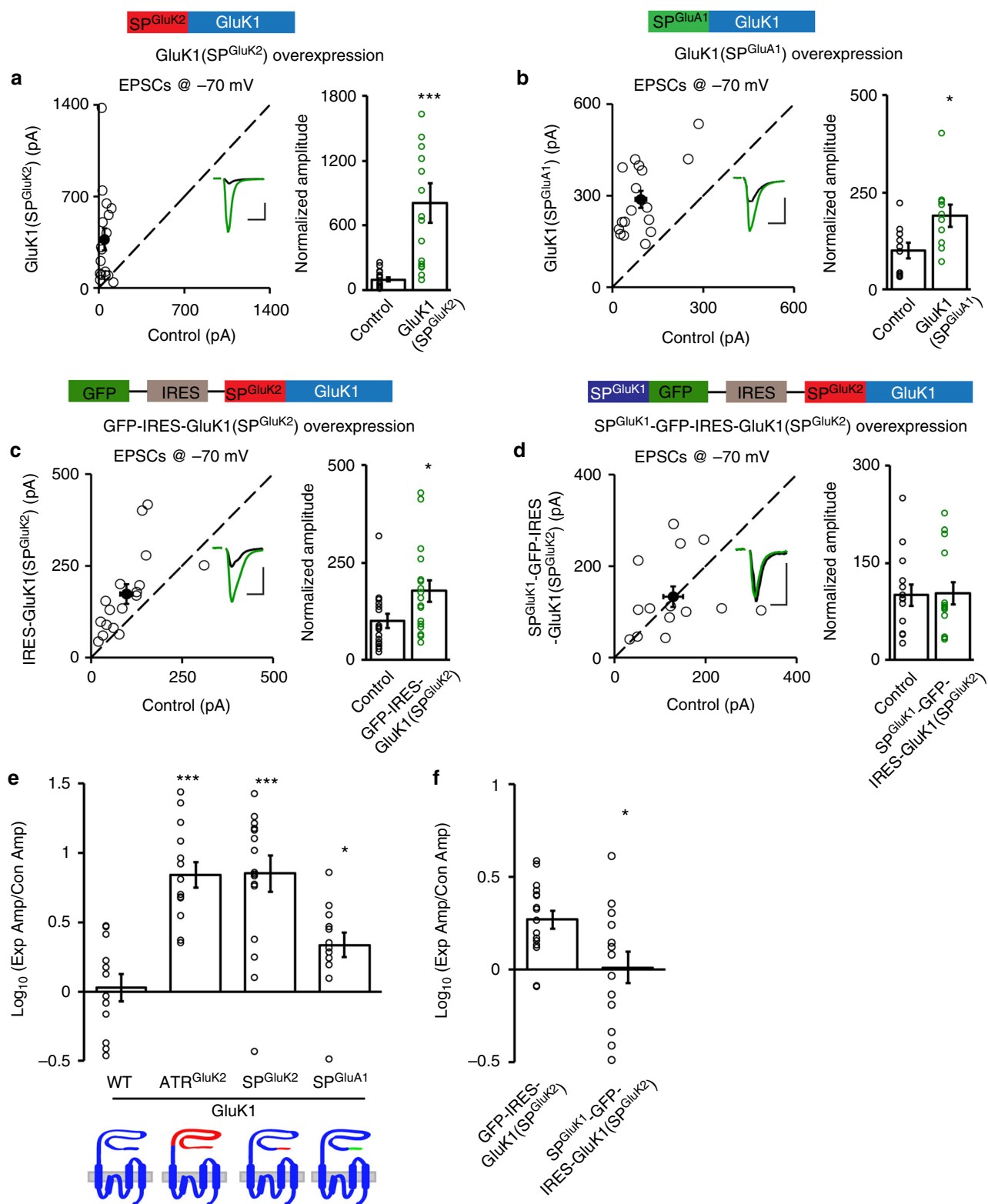

peptides was different. We first used an online program SignalP 3.0 (http://www.cbs.dtu.dk/services/SignalP-3.0/) for signal peptide prediction and found that the signal peptide cleavage probability of GluK1 is lower than that of GluK2 (Supplementary Fig. 2, GluK1: 0.532, GluK2: 0.980).

Next, we inserted an HA-tag before the signal peptide of GluK1 or GluK2 to construct HA-GluK1 or HA-GluK2 (Fig. 2a and

Supplementary Fig. 1b) and used western blot to examine their expression in HEK cells. The rationale is that if the signal peptide is not efficiently excised the protein would be detected by the HA antibody. In addition, we used SP-HA-GluK1 and SP-HA-GluK2 (Fig. 2a and Supplementary Fig. 1b) as controls, which have the HA-tag after signal peptides and should be recognized by the HA antibody. To avoid potential interference of the receptor

**Fig. 1** GluK1 signal peptide inhibits its synaptic trafficking. **a–d** Rat hippocampal slice cultures were biolistically transfected with indicated constructs and simultaneous dual whole-cell recordings from a transfected CA1 pyramidal neuron (green trace) and a neighboring wild-type one (black trace) were performed. The evoked EPSCs (eEPSCs) were measured at −70 mV. Open and filled circles represent amplitudes for single pairs and mean ± SEM, respectively. Insets show sample current traces from control (black) and experimental (green) cells. The scale bars were 200 pA/25 ms for **a** and 100 pA/25 ms for **b–d**. Bar graphs, overlaid with the actual data points, show normalized eEPSC amplitudes (mean ± SEM) presented in scatter plots (**a**, GluK1 (SP$^{GluK2}$), $n = 17$, 809.69 ± 183.98% control, ***$p < 0.001$; **b**, GluK1(SP$^{GluA1}$), $n = 14$, 189.69 ± 29.03% control, *$p < 0.05$; **c**, GFP-IRES-GluK1(SP$^{GluK2}$), $n = 17$, 177.72 ± 27.53% control, *$p < 0.05$; **d**, SP$^{GluK1}$-GFP-IRES-GluK1(SP$^{GluK2}$), $n = 14$, 103.32 ± 17.40% control, $p > 0.05$). A schematic cartoon for the construct used for transfection is shown above each graph. Statistical analyses are compared to respective control neurons with two-tailed Wilcoxon signed-rank sum test. **e** Bar graph, overlaid with the actual data points, shows logarithm summary of the eEPSC amplitude ratios (mean ± SEM) between the experimental and respective control neurons for the indicated transfections (GluK1: 0.03 ± 0.10; GluK1(ATR$^{GluK2}$): 0.84 ± 0.09; GluK1(SP$^{GluK2}$): 0.85 ± 0.13; GluK1(SP$^{GluA1}$): 0.34 ± 0.09, *$p < 0.05$, ***$p < 0.001$, compared to GluK1). **f** Bar graph shows logarithm summary of the eEPSC amplitude ratios (mean ± SEM) between the experimental and respective control neurons for the indicated transfections (GFP-IRES-GluK1(SP$^{GluK2}$): 0.31 ± 0.05; SP$^{GluK1}$-GFP-IRES-GluK1(SP$^{GluK2}$): 0.01 ± 0.08; *$p < 0.05$). It should be noted that the raw data of GluK1, and GluK1(ATR$^{GluK2}$) are re-used from our previous study[15]. The cartoons for the swapped domains between GluK1 (blue) and GluK2 (red) or GluA1 (green) proteins are shown below the graph. Data in **e** and **f** are analyzed using Mann–Whitney $U$-test

expression by the HA-tag, we applied a short linker sequence GGGGS in our constructs (Supplementary Fig. 1b). As expected, SP-HA-GluK1 and SP-HA-GluK2 were recognized by an HA antibody (lane 3 of panel 1 in Fig. 2b, c), and wild-type GluK2 and HA-GluK2 were not detected (lanes 1 and 2 of panel 1 in Fig. 2c), suggesting that the GluK2 signal peptide is fully cleaved. However, the HA antibody detected HA-GluK1, with molecular weight slightly higher than SP-HA-GluK1 (lanes 2 and 3 of panel 1 in Fig. 2b). These results suggest that either the GluK1 signal peptide is not excised, or alternatively, the HA insertion before GluK1 signal peptide interfered with its cleavage. We therefore chose an antibody recognizing the GluK1 C-terminal sequence to re-examine these GluK1 receptors and found there were two bands in the HA-GluK1 lane (lane 2 of panel 2 in Fig. 2b). The lower molecular weight band was the same size as GluK1 (SP$^{GluK2}$) (lane 4 of panel 2 in Fig. 2b), in which the signal peptide was excised, whereas the higher molecular weight band was slightly higher than both GluK1(SP$^{GluK2}$) and SP-HA-GluK1 (lane 3 of panel 2 in Fig. 2b). These results indicate that both the HA tag and the GluK1 signal peptide are intact in the higher molecular weight band of HA-GluK1. Importantly, the wild-type GluK1 only showed one band identical to GluK1(SP$^{GluK2}$) (lanes 1 and 4 of panel 2 in Fig. 2b), indicating that the signal peptide in native GluK1 is completely cleaved. Moreover, these data suggest that insertion of the HA tag before the GluK1 signal peptide partially blocks its cleavage. Consistently, when the HA tag was inserted before the signal peptide of the GluK2(SP$^{GluK1}$) chimeric receptor (Fig. 2a, c), the resultant proteins were also recognized as two bands by the antibody against the GluK2 C-terminal sequences (lane 5 of panel 2 in Fig. 2c), while there was only one band in the lane expressing HA-GluK2 (lane 2 of panel 2 in Fig. 2c).

To further understand the function and cleavage of the GluK1 signal peptide, we expressed the fusion proteins of SP$^{GluK1}$-GFP and SP$^{GluK2}$-GFP as well as HA-tagged isoforms in HEK cells (Fig. 2d). The rationale is, if the conventional ER-targeting function of the tethered signal peptide is preserved, the soluble GFP marker should be directed into ER, and subsequently, following cleavage, secreted to cell culture medium[26]. Indeed, GFP was detected in the cell culture medium in all conditions when tethered with GluK1 or GluK2 signal peptide (lanes 1–4 of panel 4 in Fig. 2e), whereas non-tethered GFP lacks the ability to be secreted (lane 5 of panel 4 in Fig. 2e). In addition, we checked the intracellular localization of the GFP moiety by confocal microscopy. GFP alone distributed in cytosol and nucleus (Fig. 2f), consistent with previous observations that

non-membrane bound GFP can freely enter the nucleus[27]. When it was tethered with GluK1 or GluK2 signal peptide, GFP was only seen in the cytoplasm (assumingly in the ER lumen) and was absent from the nucleus (Fig. 2f), indicating it is synthesized and secreted through the ER membrane system. We further found that the cleavage of the signal peptide in HA-SP$^{GluK1}$-GFP was much less efficient than that of SP$^{GluK1}$-GFP (lanes 1 and 2 of panels 1 and 2 in Fig. 2e), while N-terminal HA-tag had no influence on GluK2 signal peptide cleavage (lanes 3 and 4 of panel 1 in Fig. 2e). These findings are consistent with the KAR results (Fig. 2b, c), suggesting that the GluK1 signal peptide is indeed cleaved in mature receptors and the interruption of the tagged HA on signal peptide cleavage is specific to GluK1. Taken together, all these results demonstrate that both GluK1 and GluK2 signal peptides are functional in ER targeting and fully cleaved in the mature receptors.

**GluK1 ATD is required for signal peptide inhibitory function.** Since the signal peptide strongly inhibits GluK1 synaptic trafficking, we wondered whether this inhibitory property is general or specific to GluK1 receptor. If it has a general inhibitory effect, then the GluK2(SP$^{GluK1}$) chimeric receptor, in which the signal peptide of GluK2 is replaced with that of GluK1, should be excluded from the synapse. However, we found that this receptor still trafficked to synapses efficiently, to the same extent as that of wild-type GluK2 (Fig. 3a, e). Therefore, the inhibitory property appears to be restricted to GluK1. Since we previously found that the synaptic targeting of GluK2 was fully repressed by the GluK1 amino-terminal region[15] (Fig. 3e), we speculated that the inhibitory function of GluK1 signal peptide might require the interplay with its own ATD. Therefore, we made a series of GluK2 mutants by replacing the sub-regions of ATR with corresponding sequences in GluK1 to dissect the minimal requirement of GluK1 ATD for inhibition. We found the GluK2(N71$^{GluK1}$) chimeric receptor, the first 70 AAs of GluK2 replaced with the first 71 AAs of GluK1, still potentiated synaptic responses to a similar extent as the GluK2(SP$^{GluK1}$) receptor (Fig. 3b, e). However, further replacement of the first 136 AAs with the corresponding region in GluK1 significantly reduced the synaptic potentiation by GluK2 (Fig. 3c, e). These results lead us to propose a model in which the GluK1 signal peptide exerts its inhibitory effect through its interaction with the region between 72–137 AAs. If this is the case, replacing this region with the corresponding 71–136 AAs of GluK2 would lead to GluK1 synaptic expression. Indeed, the GluK1(N71-

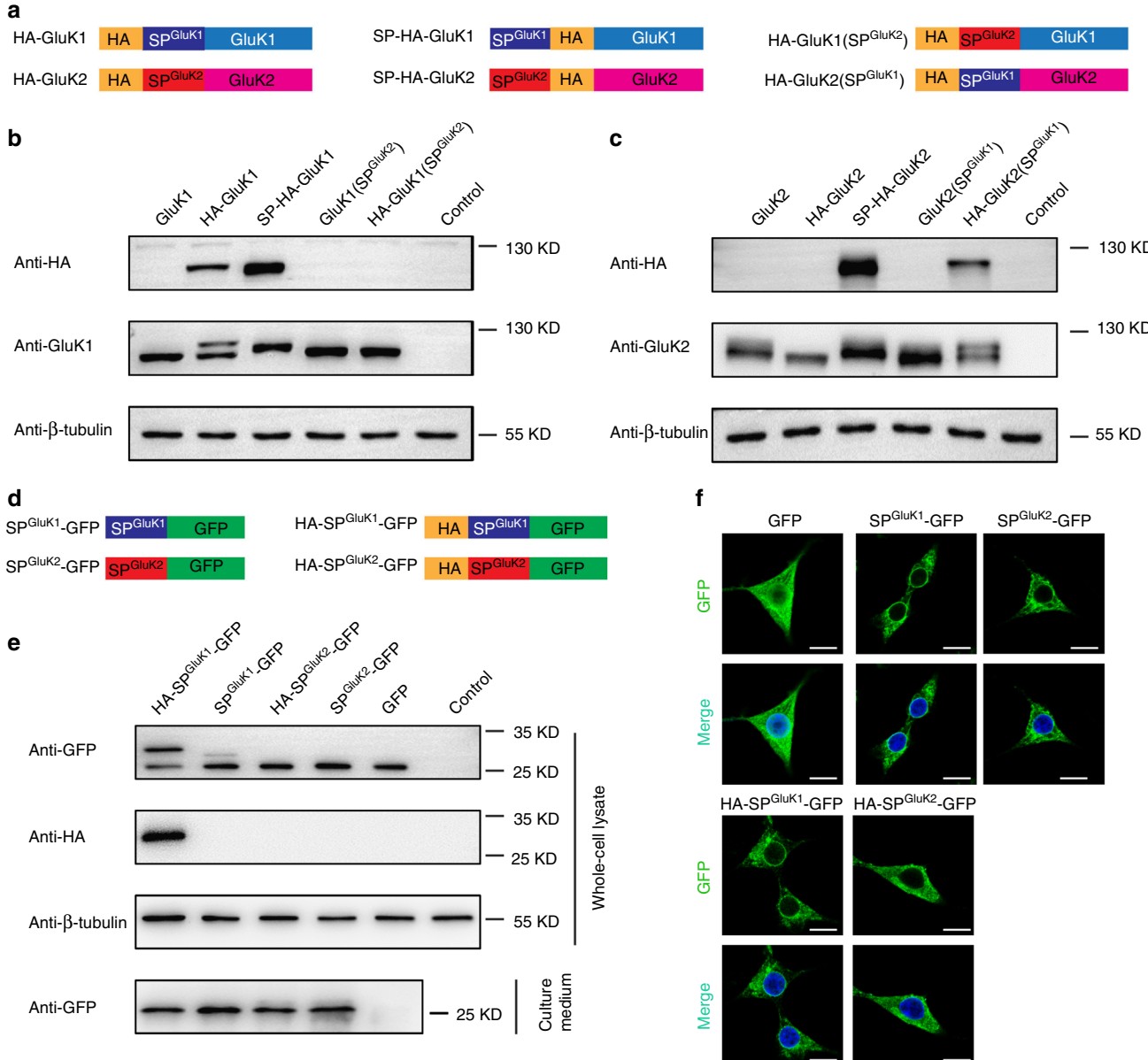

**Fig. 2** The GluK1 signal peptide is fully cleaved in the ER. **a** The schematic cartoons for the constructs of HA-tagged receptors used for transfections in **b** and **c**. **b**, **c** Immunoblot analysis of proteins extracted from HEK 293T cells with indicated transfections of wild-type and mutated GluK1 (**b**) and GluK2 (**c**) receptors. Total protein loaded was 20 μg for each sample. Panel 1, immunoblots were probed with anti-HA to detect the recombinant proteins. Panel 2, gel-shift experiment was applied and immunoblots were probed with anti-GluK1 and anti-GluK2 antibodies respectively to detect the recombinant proteins. Panel 3 was probed with anti-β-tubulin antibody as loading control. **d** The schematic cartoons for the constructs used for transfections in **e** and **f**. **e** Immunoblot analysis of the intracellular expression of tethered proteins (panel 1–3) and extracellular secreted GFP (panel 4) using indicated antibodies. **f** Intracellular localization of fluorescence signals of signal peptide-tethered GFP or non-tethered GFP in HEK 293T cells (green). The nucleus was stained with DAPI (blue). Scale bars: 10 μm

136$^{GluK2}$) chimeric receptor significantly enhanced synaptic transmission (Fig. 3d, e). We further dissected the region of 39–137 AAs in GluK1 into four sub-regions and mutated the different AAs to the corresponding ones in GluK2, none of the four mutated receptors (GluK1(M39-64), GluK1(M81-95), GluK1(M101&107), and GluK1(M116-127)) was able to traffic to the synapse (Supplementary Fig. 3a–d). These results indicate that the collaborative role of the ATD may not rely on a specific amino acid site but instead on the overall structure.

**The GluK1 signal peptide directly interacts with the ATD.** Our electrophysiological analysis demonstrated that the inhibition of GluK1 synaptic targeting required both the signal peptide and ATD, especially the region between residues 72 and 137. How does a cleaved signal peptide coordinate with the ATD to exert its inhibitory function? Might the GluK1 signal peptide directly bind to ATD? We have shown that HA insertion before the GluK1 signal peptide could impair its cleavage from the receptors or tethered GFP (Fig. 2b, c, e), and the HA-SP$^{GluK1}$-GFP fusion protein was synthesized through the ER pathway

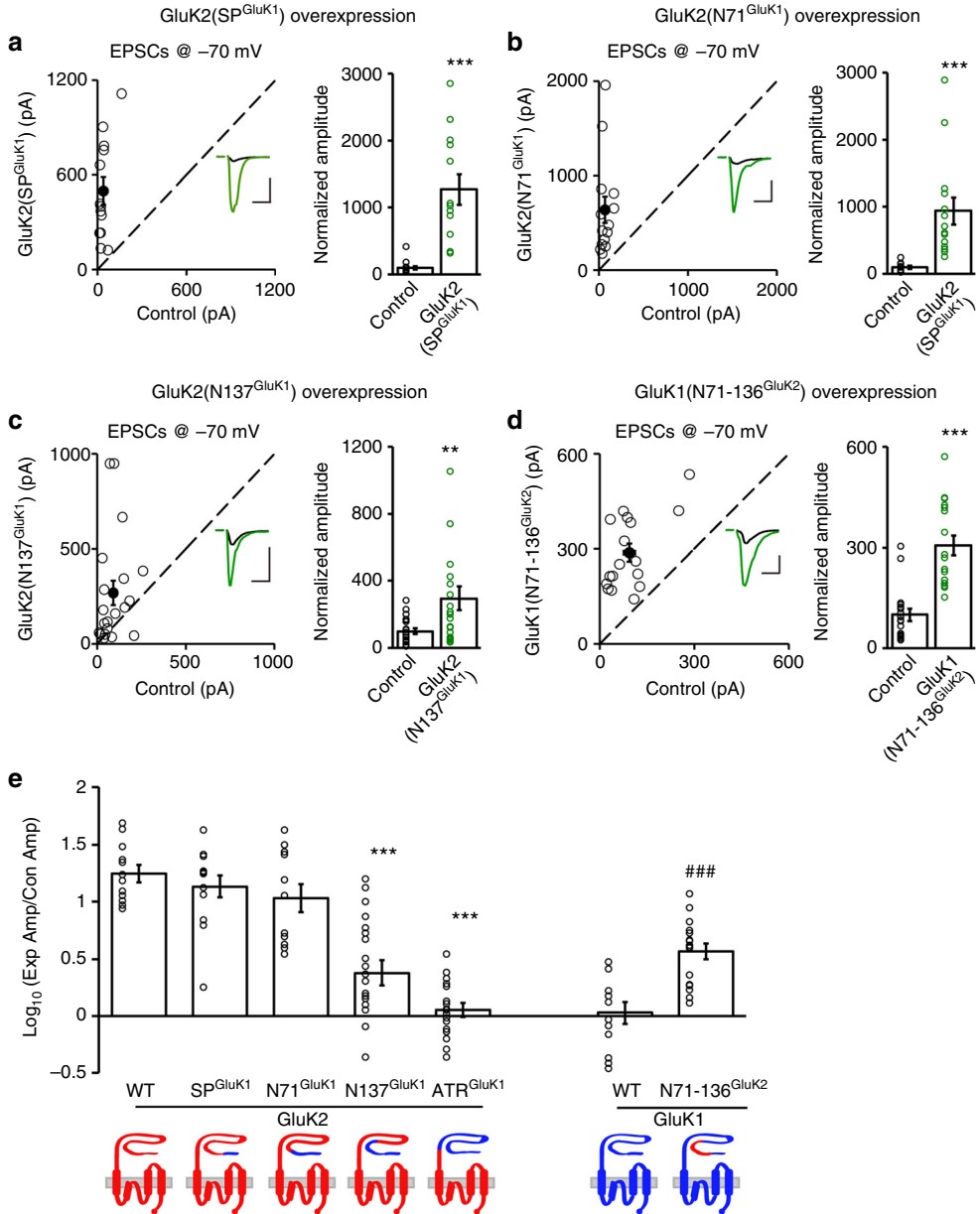

**Fig. 3** GluK1 ATD is required to restrain its synaptic targeting. **a–d** The eEPSCs measured at −70 mV for chimera receptors biolistically expressed in CA1 neurons vs. neighboring control neurons. The scale bars for representative eEPSC traces were 200 pA/25 ms for **a**, **b** and 100 pA/25 ms for **c**, **d**. Bar graphs show normalized eEPSC amplitudes (mean ± SEM) presented in scatter plots (**a** GluK2(SP$^{GluK1}$), $n = 13$, 1270.14 ± 222.55% control, ***$p < 0.0005$; **b** GluK2(N71$^{GluK1}$), $n = 14$, 939.65 ± 204.54% control, ***$p < 0.0001$; **c** GluK2(N137$^{GluK1}$), $n = 20$, 1270.14 ± 296.59% control, **$p < 0.005$; **d** GluK1(N71-136$^{GluK2}$), $n = 17$, 305.77 ± 29.94% control, ***$p < 0.0005$.) Statistical analyses are comparisons between transfected neurons and respective control neurons using two-tailed Wilcoxon signed-rank sum test. **e** Bar graph shows logarithm summary of the eEPSC amplitude ratios (mean ± SEM) between the experimental and respective control neurons for the indicated transfections (GluK2: 1.25 ± 0.07; GluK2(SP$^{GluK1}$): 1.14 ± 0.10; GluK2(N71$^{GluK1}$): 1.03 ± 0.12; GluK2(N137$^{GluK1}$): 0.38 ± 0.11; GluK2(ATR$^{GluK1}$): ***$p < 0.0001$ compared to wild-type GluK2. GluK1: 0.03 ± 0.10; GluK1(N71-136$^{GluK2}$): 0.57 ± 0.07; ###$p$ < 0.0005. It should be noted that the raw data of GluK1, GluK2 and GluK2(ATR$^{GluK1}$) are re-used from our previous study[15]. The schematic cartoons for the swapped domains between GluK1 (blue) and GluK2 (red) proteins are shown below the graph. Data in **e** are analyzed using Mann–Whitney $U$-test

(Fig. 2f). Thus, we used HA-SP$^{GluK1}$-GFP to test its binding capability to FLAG-tagged GluK1 or GluK2 ATD, in which the GluK2 signal peptide was used to ensure its expression in the ER pathway (Fig. 4a). These fusion proteins could be recognized based on their expected sizes with anti-HA or anti-FLAG antibody, respectively (left panels in Fig. 4a). After coexpression in HEK cells, HA-SP$^{GluK1}$-GFP could efficiently pull-down ATD$^{GluK1}$-FLAG, but was weakly co-immunoprecipitated with ATD$^{GluK2}$-FLAG (lanes 4 and 5 of right panel 1 in Fig. 4a). Furthermore, a control construct lacking GluK1 signal

sequence, HA-GFP failed to pull-down ATD$^{GluK1}$-FLAG or ATD$^{GluK2}$-FLAG (lanes 2 and 3 of right panel 1 in Fig. 4a). Therefore, in HEK cells, the GluK1 signal peptide appears to specifically interact with the GluK1 ATD.

One concern is that the HA-SP$^{GluK1}$-GFP construct might create additional reactive binding sites between domains. Therefore, we sought to test the direct binding of GluK1 signal peptide with its ATD in vitro. We synthesized fluorescein isothiocyanate (FITC)-labelled GluK1 signal peptide and used fluorescence polarization (FP) assays to test its binding affinity with GST-

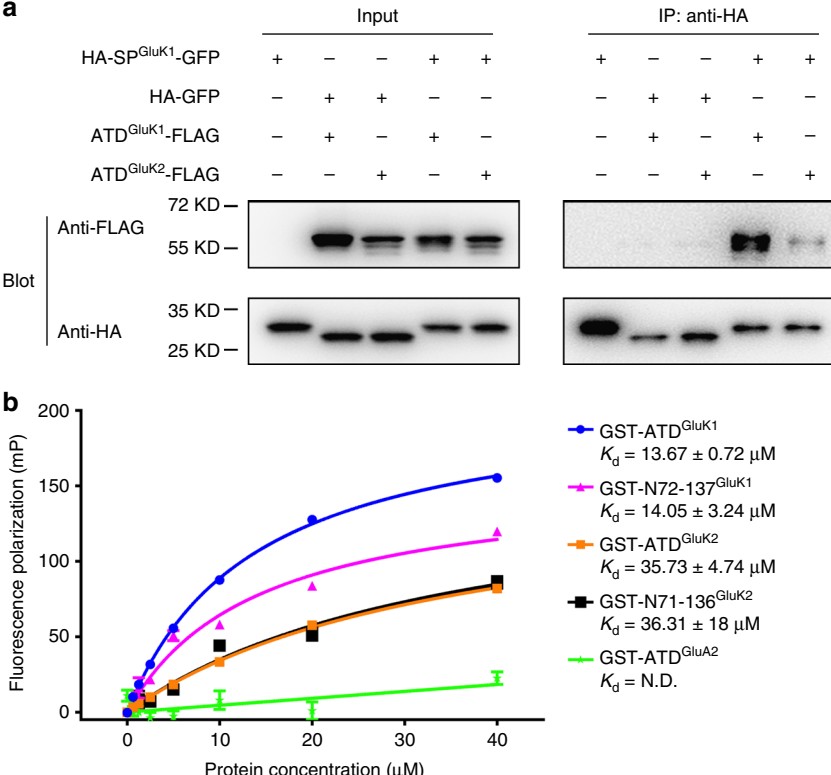

**Fig. 4** The GluK1 signal peptide interacts with the ATD. **a** Immunoblots of immunoprecipitates from transfected HEK 293T cells lysates. The transfected constructs are indicated above each lane. Recombinant protein HA-GFP was used as a control for HA-SP$^{GluK1}$-GFP, which lacks the GluK1 signal sequence in mature protein. HA-GFP, ATD$^{GluK1}$-FLAG, and ATD$^{GluK2}$-FLAG were driven by GluK2 signal peptide to ensure they were synthesized in ER pathway. **b** Fluorescence polarization (FP) analyses of peptide-protein. FITC-labeled GluK1 signal peptide (100 nM) was titrated with increasing concentrations of indicated GST-fusion proteins, and changes in FP were monitored and the apparent $K_d$ values (mean ± SEM) were determined from three experiments. N.D., not determined for GST-ATD$^{GluA2}$

tagged GluK1 ATD or the critical sub-region purified from transduced *E. coli* (Fig. 4b). Although these proteins produced by bacteria might not be folded and matured to the same extent as in eukaryotic cells, both the GluK1 ATD and N72-137 region showed saturable binding to the GluK1 signal peptide, and the affinity was significantly higher than to the corresponding regions of GluK2 (Fig. 4b). Moreover, the ATD of the AMPA receptor GluA2 subunit (GST-ATD$^{GluA2}$), did not show clear binding to GluK1 signal peptide (Fig. 4b). Taken together, these results suggest that the GluK1 signal peptide directly interacts with the ATD.

**GluK1 signal peptide inhibits neuronal surface trafficking**. We next determined whether the mechanism revealed above is involved in the surface delivery of GluK1 receptors. We therefore transfected primary cultured hippocampal neurons with the GFP expression vector together with the above critical chimeric GluK1 receptors, either GluK1(SP$^{GluK2}$) or GluK1(N71-136$^{GluK2}$), harboring an HA-tag after the signal peptide for immunostaining. The HA-tag is located extracellularly, allowing us to evaluate and quantify the surface and intracellular expression of the receptors by a two-step immunostaining protocol. The surface expression was analyzed using a mouse anti-HA antibody and a goat anti-mouse secondary antibody under membrane impermeable condition. Then the neurons were permeabilized and intracellular receptors was examined by a rabbit anti-HA antibody and a goat anti-rabbit secondary antibody. Consistent with our previous studies[14,15], although the expression levels of GFP tracer were

very similar across different conditions (Fig. 5a) and the GluK1 and GluK2 proteins were clearly expressed intracellularly in these neurons (Fig. 5a), the surface expression of GluK1 receptors was very limited, while GluK2 receptors robustly expressed at the surface (Fig. 5a, b). Moreover, the surface trafficking of GluK1 receptors was significantly increased when its signal peptide or the region between 72-137AAs was replaced with the corresponding sequences in GluK2 (Fig. 5a, b). These results suggest that the GluK1 signal peptide and the N72-137 region are also critical for the receptor surface delivery.

We have shown that the signal peptide regulates GluK1 synaptic targeting in a *trans* manner (Fig. 1). We further wondered whether this regulation was involved in regulation of neuronal surface trafficking. Using the same strategy as in Fig. 5a, we found that when SP$^{GluK1}$-GFP was coexpressed with GluK1 (SP$^{GluK2}$) in primary cultured neurons, the neuronal surface expression of GluK1(SP$^{GluK2}$) was significantly decreased (Fig. 5c, d). Taken together, these results suggest that the interaction between the signal peptide and ATD also represses GluK1 receptor surface trafficking.

## Discussion
In this study, we used hippocampal Shaffer collateral-CA1 synapses as a null-background system to study the intrinsic mechanism governing the different forward trafficking capabilities of GluK1 and GluK2 receptors. We find a repressive role of the GluK1 signal peptide during this process. Furthermore,

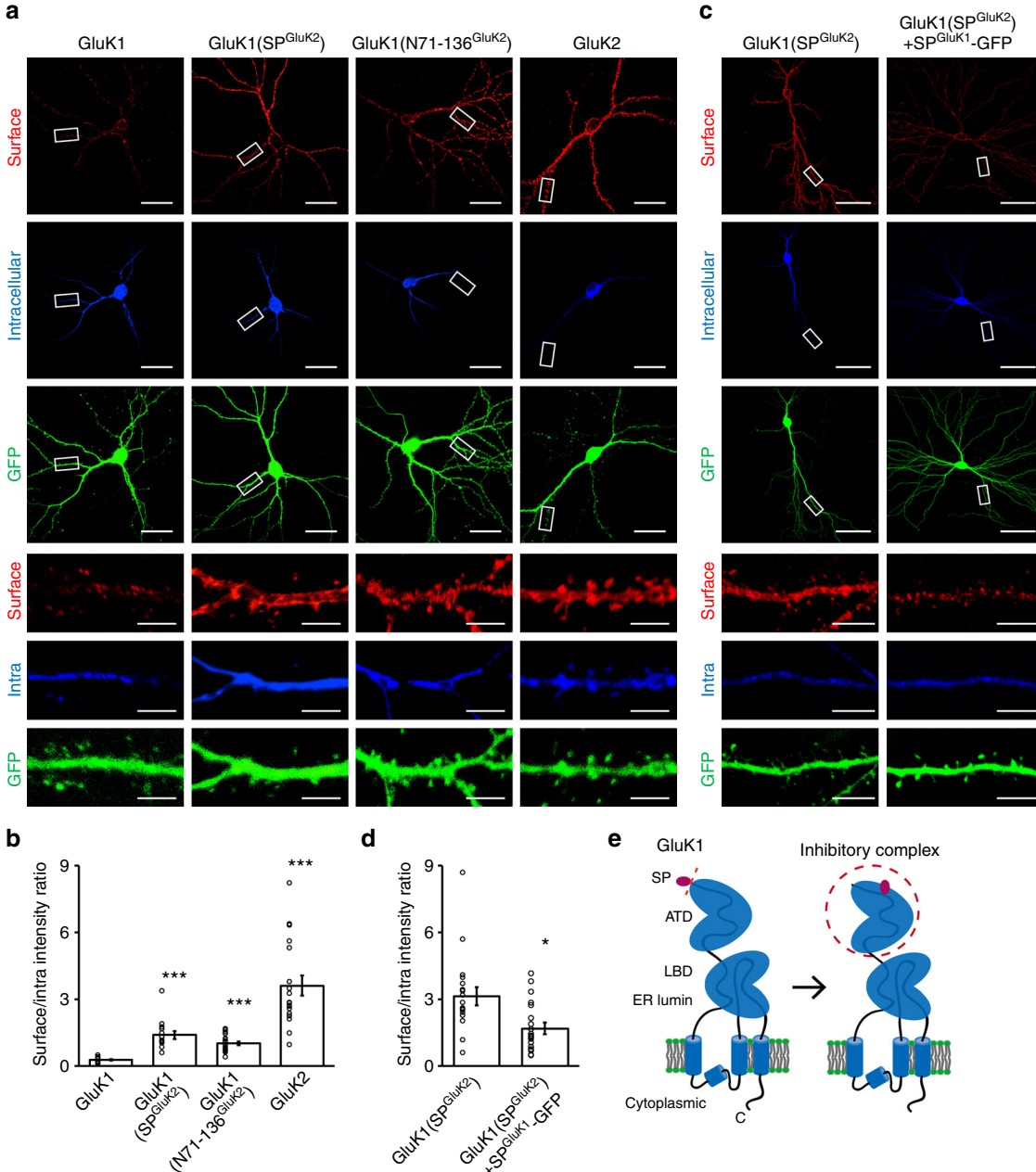

**Fig. 5** GluK1 signal peptide and ATD control its receptor surface expression. **a** Representative images from cultured hippocampal neurons transfected with wild-type or chimeric kainate receptors as indicated. An HA-tag was inserted after the signal peptide for immunostaining analysis. Neurons were transfected at DIV12 and stained at DIV16-18. Plasma membrane (red) and intracellular (blue) expressed receptors were differentially labeled before and after permeabilization respectively. pCAGGS-GFP empty vector was co-transfected and used as a volume marker to visualize neuron morphology. Expanded images at the bottom of each panel are taken from the area indicated by white rectangles. Scale bar: 25 μm (original) and 4 μm (enlarged). **b** Quantitation of the relative expression of indicated receptors at neuronal surface to the entire cell, as calculated from the ratio of red to blue fluorescence intensity. Bar graph, overlaid with the actual data points, shows the total results from three independent experiments (mean ± SEM): GluK1, $n = 17$, 0.29 ± 0.04; GluK1(SP$^{GluK2}$), $n = 14$, 1.41 ± 0.18; GluK1(N71-136$^{GluK2}$), $n = 19$, 1.02 ± 0.08; GluK2, $n = 18$, 3.63 ± 0.46; ***$p < 0.0001$, Statistical analyses are comparisons to GluK1 with Mann–Whitney $U$-test. **c** Surface and intracellular expression of GluK1(SP$^{GluK2}$) (HA-tagged after signal peptide, left panels) or SP$^{GluK1}$-GFP cotransfetion (right panels) as indicated. Expanded images at the bottom of each panel are taken from the area indicated by white rectangles. Scale bars: 50 μm (original) and 5 μm (enlarged). **d** Quantitative analysis for **c** (mean ± SEM): GluK1(SP$^{GluK2}$), $n = 19$, 3.15 ± 0.41; GluK1(SP$^{GluK2}$) + SP$^{GluK1}$-GFP, $n = 20$, 1.70 ± 0.25; *$p < 0.05$. Data are analyzed using Mann–Whitney $U$-test. **e** The working model of GluK1 showing the signal peptide and ATD forming an inhibitory complex to regulate the receptor trafficking

both the surface trafficking and synaptic targeting of GluK1 receptors are repressed by the signal peptide through direct interaction with the ATD.

Numerous studies have demonstrated that the intracellular CTD of kainate receptors are involved in their surface trafficking[17,28]. Recent studies have shown that extracellular

regions, especially the ATD, also play important roles in regulating KARs surface and synaptic expression[15,18,19]. Our previous studies have found that in CA1 neurons, GluK1 receptors have very limited surface expression and they are completely excluded from synapses, whereas GluK2 receptors express well on the neuronal surface and significantly enhances synaptic currents[14,15]. The present results further pinpoint the difference in their forward trafficking abilities to the ATD and signal peptide. A likely model for the limited trafficking capability of GluK1 is that its signal peptide, after cleavage in ER, binds to its ATD and forms an inhibitory complex that represses its surface and synaptic trafficking (Fig. 5e). By replacing either the signal peptide with that of other glutamate receptors or the ATD region with the corresponding GluK2 sequences, the inhibitory complex is disrupted and GluK1 can now appear on the neuronal surface and at the synapse. Transferring this inhibitory complex containing both the GluK1 signal peptide and the ATD to GluK2 completely eliminates GluK2 synaptic expression[15]. Additionally, GluK2 synaptic trafficking is significantly impaired by replacement of its first 136 AAs, including the signal peptide, with the corresponding region of GluK1.

Generally, signal peptides are regarded as 'cellular address codes', which enable newly synthesized non-cytosolic proteins including membrane channels to be transported to their correct destination. However, growing evidence has revealed postcleavage physiological functions for signal peptides[21,22]. Our recent work has identified a role of the GluA1 signal peptide in determining the spatial assembly of heteromeric AMPA receptors[23], which clearly functions in a *cis* manner. Here we uncover a function of the signal peptide of GluK1, which binds to the receptor like a ligand and is involved in the regulation of KAR intracellular trafficking in a *trans* manner. The signal peptide is usually released from the translocation site into the lipid bilayer after signal peptidase cleavage and spans the ER membrane like a transmembrane protein[22]. How might the GluK1 signal peptide be released from the ER membrane and then enters the ER lumen, the presumed intracellular location for its interaction with the ATD? Multiple mechanisms have been found for the translocation of signal peptides of distinct proteins, such as Semliki Forest Virus protein p62 and MHC class I molecules[22]. The signal peptides of MHC class I molecules are highly conserved. After cleavage by signal peptidase and further processing by signal peptide peptidase in the plane of ER membrane, a peptide epitope derived from the signal peptides is imported into the ER lumen by the transporter associated with antigen processing and then forms a complex with the newly synthesized nonclassical HLA-E molecules to exert its function on the cell surface[29]. It would be of interest to know whether the GluK1 signal peptide is processed in a similar manner. Additional questions are also raised. Is the full length GluK1 signal peptide responsible of the inhibitory function or is it a partial peptide derived from it? Does the binding occur in ER lumen or in Golgi apparatus? Does the inhibition require other molecules? Can the GluK1 signal peptide regulate the intracellular trafficking of GluK1/K4 or GluK1/K5 heteromeric receptors expressed in neurons? More work will be needed to determine the above underlying molecular and cellular mechanisms in future.

## Methods

**Experimental constructs**. The cDNAs of rat GluK1, rat GluK2 were used in this study. The GluK1 and GluK2 chimeric mutants made by overlapping PCR and corresponding primers (Supplementary Table 1) were subcloned into the pCAGGS vector for biolistic transfection[15]. The HA-tagged recombinant proteins and GFP fusion proteins for western blotting and imaging were generated by overlapping PCR (Vazyme Biotech, P505) and subcloned into the pCAGGS vector. The GST-tagged fusion proteins used in fluoresence plolarization assay were generated by inserting the corresponding sequences into pCold vector which has a GST tag in N-terminal. Mutant constructs were confirmed by sequencing over the entire length of the coding region.

**Electrophysiology in slice cultures**. Organotypic hippocampal slice cultures were made from P6–P8 rats. All experiments were performed in accordance with established protocols approved by the Institutional Animal Care and Use Committee of Kunming Institute of Zoology, Chinese Academy of Sciences and Model Animal Research Center, Nanjing University. Transfections were carried out on DIV 2 after culturing, using a Helios Gene Gun (Bio-Rad) with 1 μm DNA-coated gold particles. When biolistically expressing several plasmids, gold particles were coated with equal amounts of individual plasmid that expressed different fluorescent tracers. The frequency of co-expression was nearly 100%. Slices were maintained at 34 °C with media changes every other day. On DIV 8 dual whole-cell recordings of CA1 pyramidal neurons were carried out by simultaneously recording synaptic responses from a fluorescent transfected neuron and a neighboring wild-type neuron. Pyramidal neurons were identified by morphology and location. Series resistance was monitored on-line, and recordings in which the series increased to >30 MOhm or varied by >50% between neurons were discarded. Slices were superfused with artificial cerebrospinal fluid (ACSF) bubbled with 95% $O_2$/5% $CO_2$ consisting of (in mM): 119 NaCl, 2.5 KCl, 4 $CaCl_2$, 4 $MgSO_4$, 1 $NaH_2PO_4$, 26.2 $NaHCO_3$, 11 Glucose. Picrotoxin (100 μM) was added to block inhibitory currents and 2-Chloroadenosine (4 μM) was used to control epileptiform activity. The internal solution contained (in mM): 135 $CsMeSO_4$, 8 NaCl, 10 HEPES, 0.3 EGTA, 5 QX314-Cl, 4 MgATP, 0.3 $Na_3GTP$, 0.1 spermine. A bipolar stimulation electrode was placed in stratum radiatum, and responses were evoked at 0.2 Hz. Peak AMPAR and KAR currents were recorded at −70 mV. Data was analyzed off-line with custom software (IGOR Pro). Responses were collected with a Multiclamp 700A amplifier (Axon Instruments), filtered at 2 kHz, and digitized at 10 kHz.

**Surface immunolabeling and imaging**. To determine surface expression, an HA tag was inserted on wild-type and mutant kainate receptors after the signal peptide. Hippocampal neurons were prepared from E17.5, transfected at DIV 12, and fixed at DIV 16–18 with 4% paraformaldehyde supplied with 4% sucrose in phosphate-buffered saline (PBS) for 10 min at room temperature. After blocking, surface receptors (red) were labeled with a mouse anti-HA antibody (Sigma, Cat. No. H3663, 1:500) at room temperature for 2 h followed by Alexa-555 secondary antibody (Invitrogen, Cat. No. A-21422, 1:1000). Then neurons were permeabilized with blocking buffer containing 0.4% TritonX-100. The intracellular receptors were immunostained with a rabbit anti-HA antibody (Cell signaling technology, Cat. No. H3724, 1:250) followed by Alexa-647 secondary antibody (Invitrogen, Cat. No. A27040, 1:1000). Images were acquired by confocal microscopy (EZ-C1, 100x oil or Zeiss LSM880, 63x oil) and analyzed with ImageJ software. Each group of neurons used for comparison were imaged with the same acquisition parameters. Background-subtracted and maximum projection of Z-stack images were used for integrated intensity quantification and the intensity ratios were obtained from the surface-labeled and intracellularly labeled channels. Fourteen to 20 neurons were measured in each group.

**Immunoprecipitation**. HEK 293T cells (ATCC) were cotransfected with indicated expression plasmids combination (1:1) in 10 cm dishes 36 h before use. Cells were washed three times with PBS and harvested and solubilized in lysis buffer (50 mM Tris-Cl (pH 7.2), 150 mM NaCl, 2 mM EDTA, and 0.1% Triton X-100), supplemented with a mixture of protease inhibitors (Roche) and solubilized for 1 h at 4 °C. After centrifuged at 13,800 × g for 20 min, the pellet was discarded. Lysates were then incubated with antibodies at 4 °C overnight. Lysates were then incubated with Protein G beads (GE Healthcare) for 2 h at 4 °C on a rotating platform. After incubation, beads were washed four times with lysis buffer and boiled in 40 μl 2×Laemmli buffer.

**Western blots**. HEK 293T cells were transiently transfected using Lipofectamine 2000 reagent (Invitrogen) following the manufacturer's instructions. After 36 h, cells were lysed in RIPA buffer containing 150 mM NaCl, 50 mM Tris (pH 7.4), 1% Nonidet P-40, 0.5% sodium deoxycholate, and a mixture of protease inhibitors (Roche). After incubation on ice for 30 min, cell lysates were centrifuged for 30 min at 13,800 × g at 4 °C. Then the supernatant was mixed with 5× loading buffer and DTT and boiled. For detection of full-length receptors, the mixture was immediately loaded onto 6% SDS-PAGE gels. The protein bands were transferred to PVDF membranes (Millipore) at 100 V for 2 h, and then blocked in 5% non-fat milk dissolved in TBST at room temperature for 1 h. Finally, the level of GluK1 and GluK2 receptors were probed with anti-HA antibody (Sigma, Cat. No. H3663, 1:2000), anti-C-terminal GluR6/7 (Merk Millipore, Cat. No. 04–921, 1:8000), and anti-C-terminal GluR5 (Merk Millipore, Cat. No. 07–258, 1:1000) respectively, and detected using the ECL substrate (Thermo) before exposure. For detection of GFP fusion proteins, the mixture was immediately loaded onto 10% SDS-PAGE gels and probed with rabbit anti-GFP antibody (MBL, Cat. No. 598, 1:2000) and anti-HA antibody (Sigma, Cat. No. H3663, 1:2000).

**Detection of the secreted proteins.** Secreted GFP fusion proteins were detected by immunoblotting. HEK 293T cells were cultured and transiently transfected with the indicated constructs and after 48 h the culture medium was collected and cell debris was removed by centrifugation ($1000 \times g$, 3 min). The culture medium was then incubated with rabbit anti-GFP antibody at 4 °C overnight. Then the medium was incubated with Protein G beads (GE Healthcare) for 2 h at 4 °C on a rotating platform. After incubation, beads were washed four times with PBS and boiled in 50 μl 2×Laemmli buffer. The mixture was then centrifuged at $13,800 \times g$ and the supernatant was used for western blotting.

**Immunocytochemistry and confocal microscopy.** GFP fusion proteins were detected by immunocytochemistry. HEK293T cells were washed in PBS and fixed in 4% PFA in PBS. After blocking in normal goat serum, GFP staining was examined using rabbit anti-GFP antibody (MBL, Cat. No. 598, 1:500), followed by Alexa-488 secondary antibody (Invitrogen, Cat. No. A-11034, 1:1000). After the secondary antibody was washed by PBS for three times, cells were additionally incubated with DAPI for nuclear staining. Samples were examined and analyzed through a 63 X oil immersion lens on a Zeiss LSM880 microscope.

**Production of GST fusion proteins.** GST fusion proteins were expressed and purified from BL21(DE3) *E. coli* (Vazyme Biotech). Briefly, transformed bacteria were incubated with 3 L LB medium at 37 °C until the cultures reached an $OD_{600}$ of 1.0. The cultures were then induced at 16 °C with 1 mM IPTG in an ice-water bath for 30 min. Bacteria were harvested by centrifugation and resuspended in 50 mL PBS, supplemented with 1 mM PMSF and protease cocktail (Roche), followed by sonication. Lysates were clarified by centrifugation and incubated with glutathione Sepharose 4B (Amersham). Finally, proteins were washed with PBS and eluted with elution buffer (10 mM glutathione, 150 mM NaCl, 50 mM Tris (pH 8.0)) and concentrated by ultrafiltration using Amicon Ultra-15 column (Millipore). The purity and quality of the fusion proteins was evaluated by SDS-polyacrylamide gel electrophoresis and detection by staining with Coomassie Brilliant Blue and by immunoblotting with anti-GST antibody to ensure minimal degradation.

**Fluorescence polarization assay.** Saturation binding curves for protein-peptide interactions were determined by examining the change in fluorescence polarization (FP). The GluK1 signal peptide was synthesized with a fluorescein isothiocyanate (FITC)-labeled at N-terminal. Peptide probes (100 nM) were used in the titration with increasing amount of GST-fusion proteins in 100 μl FP buffer (100 mM KCl, 20 mM HEPES (pH 7.4)) in 96-well black polystyrene plates (Corning). The plate was read on a Synergy H1 microplate reader (BioTek) in the fluorescence polarimeter mode to acquire polarization values ($P$). The resulting $P$ values were plotted against the concentration of proteins after subtraction of baseline. To determine the $K_d$ value, a curve was fitted by the equation $Y = B * X/(K_d + X)$, with $B$ being the maximum $P$ value that would be reached at saturation as indicated by the extrapolation of the fitted curve.

**Statistical analysis.** Significance of the difference between responses in the transfected cells compared to the control cells was determined using the two-tailed Wilcoxon signed-rank sum test. For all experiments involving un-paired data, a Mann–Whitney $U$-test with Bonferonni correction for multiple comparisons was used. Data analysis was carried out in Igor Pro (Wavemetrics), Excel (Microsoft), and GraphPad Prism (GraphPad Software).

**Reporting Summary.** Further information on research design is available in the Nature Research Reporting Summary linked to this article.

## Data availability
Data supporting the findings of this manuscript are available from the corresponding authors upon reasonable request. A reporting summary for this Article is available as a Supplementary Information File.

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

## Acknowledgements

We thank Dr. Jiajia Liu (Institute of Genetics and Developmental Biology, Chinese Academy of Sciences) for technical advice in imaging of cultured neurons. This work was supported by grants the Strategic Priority Research Program of the Chinese Academy of Sciences (XDB13000000), the Ministry of Science and Technology of China (2014CB942804 and 2015BAI08B02), National Natural Science Foundation of China (31741055, 31571060 and 31371061), the Chinese Academy of Sciences Pioneer Hundred

Talents Program (to N.S.), the Foundation of State Key Laboratory of Genetic Resources and Evolution, Kunming Institute of Zoology Chinese Academy of Sciences (GREKF18-19), the Fundamental Research Funds for the Central Universities (0903-14380021), and Natural Science Foundation of Jiangsu Province Grant (BK20140018). R.A.N is supported by grants from NIMH, NIH. We are grateful to all members of the Sheng and Shi laboratory for their discussion of and comments on the manuscript.

## Author contributions

Y.S.S., N.S., and R.A.N. designed the experiments. N.S., G.F.D., Y.Y., S.X., W.T., S.Z., and T.J. performed the experiments. N.S., G.F.D., and T.J. analyzed data. Y.S.S. and N.S. wrote the paper.

## Additional information

**Competing interests:** The authors declare no competing interests.

