## [Peer Review File · Nature Communications]

Reviewers' Comments:

Reviewer #1:

Remarks to the Author:

In their manuscript, Duan and coworkers describe the repression of the kainate receptor GluK1 trafficking by its own cleavable signal peptide. They propose that the cleaved signal peptide interacts with a region of the ATD of the mature receptor, namely between residues 72 and 137. If this holds true, this represents a highly interesting novel mechanism for a "post ER" function of a signal peptide which would not only explain the different trafficking properties of GluK1 and GluK2, these data would also be relevant for protein trafficking in general. Whereas the majority of the experiments were carried out carefully and are convincing, I have a major concern addressing the design and results of the Co-IP experiments (see below).

Minor points

1. The electrophysiological experiments are convincing and also those showing that the signal peptide of GluK1 acts in trans (Fig 1). The immunodetection experiments for SP cleavage of GluK1 described in Fig. 2 are, however, difficult to interpret, mainly because the HA tag obviously impairs cleavage of the SP of GluK1, as mentioned by the authors. The verification of cleavage then relies only on very small size differences of the apparent molecular masses seen on the anti-GluK1 panel.

While I think that the overall interpretation of the bands of Fig. 2 might be correct, the authors could perform a simple experiment to prove cleavage unambiguously (and also somewhat quantitatively). They could take their SP(GluK1)-GFP construct, express it transiently in cells, and monitor the fate of the GFP moiety. In case of a functional and cleaved SP, the SP converts GFP into a secretory protein and the GFP moiety is translocated to the ER lumen via the Sec61 complex. Following vesicular trafficking, it is released into the cell culture medium where it can be detected either fluorimetrically or by immunoblotting. The authors could use their equivalent construct with impaired cleavage (HA tag) as a negative control and also a construct using the SP of GluK2 as a positive control. Such experiments were previously published by others (e.g. Schulz et al., *J. Biol. Chem.* 2010, 285:32878-32878; Vezzoli et al., *Hum. Mol. Genet.* 2015, 24:6003-6012). I recommend such an experiment, in particular because the Signal P prediction program reveals a "bad" SP in the case of GluK1 with a cleavage probability of 0.5 which is close to the cut-off value of the program (Figure S2).

2. Based on their data, the authors propose in their discussion a model where the SP of GluK1 binds to the ATD and thereby inhibits trafficking of the receptor (Fig. 4C). This is not a matter of course. In order to do so, the SP must be translocated somehow from the bilayer of the ER membrane to the ER lumen. The authors should discuss possible mechanisms for this translocation. To my knowledge, two mechanisms are currently described which may move signal sequences to the ER lumen (MHC class 1 signal peptide and the signal peptide of the Semliki forest protein P62; both reviewed in Kapp et al., 2009, Link: <https://www.ncbi.nlm.nih.gov/books/NBK6322/>). Since the P62 mechanism involves multiple autoproteolysis steps, a MHC-like pathway seems to be more feasible for discussion.

Major point

1. To prove the direct interaction between the SP of GluK1 and its ATD, the authors have chosen a Co-IP approach using the constructs HA-SP(GluK1)-GFP-TM1 and SP(GluK2)-ATD(GluK1)-FLAG-TM1 (Fig. 4F). For me, the rationale of this experiment is unclear. The authors have shown before that the HA tag prevents cleavage of the SP of GluK1. They then add the HA tag to guarantee that the SP of GluK1 and GFP remain connected in the first construct for detection. In my opinion, the constructs should then have the topology shown in my attached cartoon (it would also be good to include the author's ideas for the topology of their constructs in Fig. 4F).

My point is: if the HA-SP(GluK1) remains uncleaved, it has no chance to escape the ER membrane and to interact with the ATD of the second construct in trans (?). The authors should address this issue. To strengthen these results, the authors should also use residues 72 to 137 instead of the whole ATD in their second construct. A more straightforward approach to demonstrate this direct interaction may include purification of the SP and the relevant part of the ATD of GluK1 and to perform biacore analyses. Instead, suitable crosslinking studies may also be taken into account.

Reviewer #2:

Remarks to the Author:

This is a nice paper following on from an important theme - unexpected complexity of the molecular biology of glutamate receptor trafficking.

The introduction is clearly written and introduces the topic well - however, I was surprised that the authors did not mention their own work on the AMPA receptor signal peptide until much later. I knew the work and therefore this paper made a lot of sense. However, a reader that did not know the existing finding that the SP can determine subunit dependent assembly in a related receptor could well be more resistant to this expansion of the canonical SP role.

Overall the experiments are well designed and give considerable insight. However, the model put forward is quite challenging and therefore the authors should either provide more evidence, or scale back their conclusions and interpretation and be a bit more cautious, and identify more explicitly some of the (few) weak points in their interpretation.

Major points

I think the claim of a direct interaction between the cleaved SP and the ATD controlling trafficking, lacks for direct support. For example, the biochemical results in Figure 3F are really excellent, but they are on constructs that don't represent the real receptor and that might have rather strange properties. For example, the single TM domain of a receptor with multiple TM domains is not really guaranteed to be found in the membrane in the presumed manner. For these kind of "display" experiments, it would have been better to use a known single TM helix, like from Neto or pDisplay or similar. The authors could acknowledge this.

The key co-immunoprecipitation experiment is different from the model the authors present. The experimental interaction here is of the non-cleaved SP (hindered with HA) attached to a membrane protein, with the ATD. That's rather different from the free SP (which has a different fate following processing) - and we don't know where this interaction (which compartment) is happening in this artificial system. Very difficult to assess, but for this reason, the conclusions must be more measured. The coIP doesn't, as the authors claim, show that "the GluK1 signal peptide directly interacts with the ATD and thereby represses GluK1 trafficking." - only that the hindered SP, when tethered to the membrane, can coIP with some part of the SPGluK2ATDGluK1-Flag-TM1 construct. Some parts of this construct may not be inert. This experiment would be much stronger if controlled against the K2 ATD.

The inhibition in trans (a very impressive result from Figure 1) brings an important point forward. To be certain that the effect of the trans-inhibition is on trafficking, it would be important to demonstrate that the surface expression was reduced in this experiment. It's conceivable that the trans expression stops synaptic incorporation, without doing much to surface expression. This would be a further confounding element. Fusion of the GluK1 SP to mScarlet or a similar would allow imaging experiments to be redone with GFP tracer. In principle, the inhibitory SP could then be visualized at intracellular sites (again very challenging). Maybe just acknowledging this possibility is enough.

There is not enough information about the sequences used. The authors say that they inserted the HA tag before the signal peptide (p6). It would be important to see the exact sequence. It seems hard to do this without disrupting the SP itself but perhaps sharing more of the sequences would make it clear.

Minor

There are numerous grammar mistakes: e.g. "The GluK1 signal peptide and ATD are interacted directly" that could be fixed. A native speaker (for example, the prepenultimate coauthor; other authors may also be native speakers) could help with this.

Andrew Plested

Reviewer #3:

Remarks to the Author:

The manuscript by Duan and colleagues examines the role of the kainate receptor (KAR) signal peptide in regulating the targeting of GluK1 and GluK2 KA-type ionotropic glutamate receptors to synapses of CA1 hippocampal neurons. Previous work from the senior author has shown that GluK2 receptors readily target to the Schaffer collateral-CA1 synapse in a manner independent of the auxiliary Neto proteins, Neto1 and Neto2. In contrast, GluK1 expression is much more limited and relies almost exclusively on Neto1 and Neto2 for trafficking and synaptic targeting (Sheng et al, 2015, Elife). A follow-up study then showed that this distinction between GluK1 and GluK2 is dependent on the amino-terminal region of each subunit ATD (Sheng et al, 2016, PNAS) though the exact mechanism by which the ATD region mediates this effect was not identified. In the present study, the authors attempt to dissect apart this mechanism(s) by looking more closely at the respective signal peptides of each subunit. The authors postulate a non-canonical role for the signal peptide of GluK1 which they propose directly interacts with the ATD region to inhibit the surface expression and synaptic targeting of the GluK1 receptor.

Although the study is of potential interest to neuroscientists interested in the biochemical make-up of central synapses, the manuscript suffers from several weaknesses that have tempered my overall enthusiasm for it. First, the authors use 3 different expression systems (organotypic cultures, primary cultured neurons and finally HEK 293 cells) displayed in 4 figures to study the role of the signal peptide. The underlying assumption is that the biochemical processes that regulate GluK1 trafficking/targeting are the same in each model system, however, there are many reasons why this might not be the case. Consequently, the authors need to document why switching between 3 very different model systems is valid - ideally by showing the same primary observations in all 3 systems. Second, in terms of novelty, the authors' study is more like an extension of the previous published studies (in Elife, PNAS and JBC) looking at KAR trafficking. The unexpected role of the signal peptide is novel, however, as discussed below, the authors need a stronger dataset to make this point. Third, the text of the manuscript is very unclear in many sections which made it difficult to fully understand the authors' arguments/rationale. The manuscript would benefit from a careful review of the text to improve clarity for the reader. I have included other major concerns below to help the authors in the revision of their work.

Major

1. Manuscript text. The text in general has a lot of awkward language and/or poor sentence construction. For example, the following sentence from the "Introduction" I think should read, "Both surface (EXPRESSION) and synaptic trafficking of (THE) GluK1 receptor fully relies on auxiliary Neto proteins, while GluK2 itself traffics to the surface and(/OR THE) synapse independent of Neto proteins ". I found similar problems in many areas of the Results section of

the manuscript which made it difficult to read. For example, I am still not clear what the authors are referring to in this sentence from the Results section, "Therefore, we coexpressed the GluK1 signal peptide together with GluK1(SP2) in the same neuron and then examine the effect on synaptic transmission. To this end, we put GluK1(SP2) after IRES to reduce its expression and found this GFP-IRES-GluK1(SP2) construct still significantly increased synaptic responses in CA1 cells (Fig. 1C)". The authors need to revise the entire text of the manuscript thoroughly.

2. Model system. CA1 neurons express KARs (Castillo et al, 1997; Bureau et al, 1999) but not at the Schaffer collateral synapse which begs the question why they are not targeted to this synapse. The authors have shown that they can achieve this but it requires the overexpression of GluK1 (with Netos) and GluK2. Consequently, it could be concluded that the authors are studying an assembly mechanism that only occurs when the KAR proteins are expressed in abundance – which apparently isn't the case for wildtype CA1 neurons. Also, it seems unlikely that the mechanism responsible for endogenous KAR expression/targeting is due to the signal peptide so what might be going on? The authors need to address these confounding factors since they undermine the conclusions that can be drawn from the authors' work. Also, how do the authors know that GluK4 and GluK5 receptors are not contributing to the responses observed in the CA1 neurons? GluK4 and GluK5 only reach the cell's surface when co-expressed with GluK1,2 or 3. Consequently it is possible that the receptors may be heteromeric in nature – possibly containing GluK4/5.

3. Control cell recordings. The authors record from both transfected and non-transfected in the same organotypic slice preparations to compare their responsiveness. It would be expected that the responses from untransfected or control cells would be constant across all the conditions in the manuscript. However, a perusal of the scatter plots in figure 1 and 3 suggests that the response amplitude is quite variable. For example, a comparison of the data shown in Figure 1A suggests that the control KAR responses are less than about 100pA however in other panels (e.g. 3c and 3d), the responses can be up to 300 pA. The authors need to explain the variability in the control responses. One possibility would be to show (perhaps in a supplemental figure) a statistical comparison of the control responses across all the conditions. Given the nature of synaptic transmission, the response amplitude is probably not going to exhibit a normal distribution. As it stands, the variability in the control responses adds further to the uncertainty of the authors' findings.

4. Figure 1. The text describing figure 1 was quite confusing especially in the section referring to the constructs GFP-IRES-GluK1(SP2) and SP(GluK1)-GFP-IRES- GluK1(SP2). I think I eventually understood the authors' rationale but it took me a lot of time to work it out. Clearer language would be helpful here and in other sections of the manuscript.

5. Figure 2. The authors use HEK293 cells to argue that the signal peptide of GluK1 is not degraded as would be expected from conventional cell biology work. Although the data is consistent with this argument, it still does not address whether this occurs in CA1 neurons of the hippocampus. Since this point is critical to the overall argument developed by the authors, proof that the signal peptide of GluK1 is "spared" in native cells is needed.

6. Intracellular staining. The authors argue that data in figure 4 demonstrates that different constructs differentially distribute between the intracellular compartments and plasma membrane. Although the staining pattern is clearly different between some of the constructs, I think it is very difficult to conclude that the staining which appears to be mainly intracellular is not on the cell's surface. Likewise, the appearance of surface expression can only be demonstrated conclusively by TIRF microscopy and, ideally, using a pH sensitive GFP-tagged receptor to establish that the KAR is on the surface of the plasma membrane.

Minor Comments

1. Introduction. The points raised in the first few sentences of the text need to be referenced.

Response to Reviewer #1's comments:

In their manuscript, Duan and coworkers describe the repression of the kainate receptor GluK1 trafficking by its own cleavable signal peptide. They propose that the cleaved signal peptide interacts with a region of the ATD of the mature receptor, namely between residues 72 and 137. If this holds true, this represents a highly interesting novel mechanism for a “post ER” function of a signal peptide which would not only explain the different trafficking properties of GluK1 and GluK2, these data would also be relevant for protein trafficking in general. Whereas the majority of the experiments were carried out carefully and are convincing, I have a major concern addressing the design and results of the Co-IP experiments (see below).

Minor points

1. The electrophysiological experiments are convincing and also those showing that the signal peptide of GluK1 acts in trans (Fig 1). The immunodetection experiments for SP cleavage of GluK1 described in Fig. 2 are, however, difficult to interpret, mainly because the HA tag obviously impairs cleavage of the SP of GluK1, as mentioned by the authors. The verification of cleavage then relies only on very small size differences of the apparent molecular masses seen on the anti-GluK1 panel.

While I think that the overall interpretation of the bands of Fig. 2 might be correct, the authors could perform a simple experiment to prove cleavage unambiguously (and also somewhat quantitatively). They could take their SP(GluK1)-GFP construct, express it transiently in cells, and monitor the fate of the GFP moiety. In case of a functional and cleaved SP, the SP converts GFP into a secretory protein and the GFP moiety is translocated to the ER lumen via the Sec61 complex. Following vesicular trafficking, it is released into the cell culture medium where it can be detected either fluorimetrically or by immunoblotting. The authors could use their equivalent construct with impaired cleavage (HA tag) as a negative control and also a construct using the SP of GluK2 as a positive control. Such experiments were previously published by others (e.g. Schulz et al., J. Biol. Chem. 2010, 285:32878-32878; Vezzoli et al., Hum. Mol. Genet. 2015, 24:6003-6012). I recommend such an experiment, in particular because the Signal P prediction program reveals a “bad” SP in the case of GluK1 with a cleavage probability of 0.5 which is close to the cut-off value of the program (Figure S2).

Response: We thank the reviewer for this great suggestion and have carried out the suggested experiments. We transfected HEK cells with expression plasmids of SP^{GluK1}- or SP^{GluK2} fused GFP, as well as N-terminal HA-tagged isoforms (Fig. 2d). All those constructs (even with HA-tag before signal sequences) were able to convert GFP into a secretory protein as GFP was restricted in cytosol membrane system but not in nucleus, and also appeared in culture medium (Fig. 2e and 2f), revealing the conventional ER-targeting function for these signal sequences. Western blot of the whole-cell lysates

showed the GluK1 signal peptide was partially cleaved while GluK2 SP was fully cleaved in the N-terminal HA-tagged fusion proteins (lane 1 and 2 of panel 1 in Fig. 2e), consistent with the results of SP cleavage experiments of GluK1 or GluK2 chimera receptors (Fig. 2b). We noticed a weak band higher than GFP in SP^{GluK1}-GFP (lane 2 of panel 1 in Fig. 2e), indicating incomplete cleavage. We think this is in consistent with SignalP prediction result of lower cleavage probability for GluK1 SP (Fig. S2). However, such uncleaved band was not seen with GluK1 (lane 1 of panel 2 in Fig. 2a), suggesting the cleavage is complete in receptors. We attached the results of suggested experiments down here for easy reviewing.

Figure 2d-f. the conventional ER targeting function of GluK1/K2 signal peptides.

2. Based on their data, the authors propose in their discussion a model where the SP of GluK1 binds to the ATD and thereby inhibits trafficking of the receptor (Fig. 4C). This is not a matter of course. In order to do so, the SP must be translocated somehow from the bilayer of the ER membrane to the ER lumen. The authors should discuss possible mechanisms for this translocation. To my knowledge, two mechanisms are currently described which may move signal sequences to the ER lumen (MHC class 1 signal peptide and the signal peptide of the Semliki forest protein P62; both reviewed in Kapp et al., 2009, Link: <https://www.ncbi.nlm.nih.gov/books/NBK6322/>). Since the P62 mechanism involves multiple autoproteolysis steps, a MHC-like pathway seems to be more feasible for discussion.

Response: This is a very thoughtful insight, thanks. We have added this part in discussion in the corresponding paragraph on page 15-16. We understand the real translocation procedure of GluK1 signal peptide could be much more complicated, but we wish to keep our simplified carton model in order to help readers to catch our point

(Fig. 5e).

Major point

1. To prove the direct interaction between the SP of GluK1 and its ATD, the authors have chosen a Co-IP approach using the constructs HA-SP(GluK1)-GFP-TM1 and SP(GluK2)-ATD(GluK1)-FLAG-TM1 (Fig. 4F). For me, the rationale of this experiment is unclear. The authors have shown before that the HA tag prevents cleavage of the SP of GluK1. They then add the HA tag to guarantee that the SP of GluK1 and GFP remain connected in the first construct for detection. In my opinion, the constructs should then have the topology shown in my attached cartoon (it would also be good to include the author's ideas for the topology of their constructs in Fig. 4F).

My point is: if the HA-SP(GluK1) remains uncleaved, it has no chance to escape the ER membrane and to interact with the ATD of the second construct in trans (?). The authors should address this issue. To strengthen these results, the authors should also use residues 72 to 137 instead of the whole ATD in their second construct. A more straightforward approach to demonstrate this direct interaction may include purification of the SP and the relevant part of the ATD of GluK1 and to perform biacore analyses. Instead, suitable crosslinking studies may also be taken into account.

Response: We deeply appreciate these thoughtful comments, which are very helpful for our revised experimental design. Thanks a lot for the reviewer's cartoon topology. We fully agree with the reviewer that the uncleaved HA-SP^{GluK1}-GFP could unlikely escape from ER membrane, as it was completely absent in culture medium (only one band showing excised GFP) (see above Fig 2e). However, even with this drawback in the rationale in experimental design, we did observe a clear different ability between HA-SP^{GluK1}-GFP and a control HA-GFP in pulling-down ATD^{GluK1} (Fig. 4a). The only difference between these two constructs is that the latter just lacks the GluK1 signal sequence (Fig. 4a). We do not fully understand how the signal sequence hidden in ER membrane interacts with assumedly soluble ATD in second constructs. One possible explanation is that the signal sequence exposed during cell lysis. We still present this observation in our MS, but we are open to pull back these data if the reviewer believes it is inappropriate. Moreover, please note that, under the helpful suggestion by the second reviewer, we realized the TM1 domain is unnecessary in our experiments. So all the Co-IP experiments now we carried out are without TM1 domain, which should theoretically reduce the chances of forming unexpected reactive binding site(s).

We did not carry out the experiments with residues 72 to 137, which is very small in molecular weight and impractical for western blot because of technical limitation.

The direct interaction experiments were suggested by both reviewer 1 and reviewer 2,

which make much more sense than above Co-IP experiments. We managed to examine the direct interaction of GluK1 signal peptide with its ATD or N72-137 region using fluorescence polarization (FP) assay under the help of Dr. Jin, who is now included in our author list. We analyzed the binding affinity of FITC-labelled GluK1 signal peptide with the purified GST-tagged proteins as indicated (Fig. 4b). Although GluK1 SP showed saturable binding to both GluK1 and GluK2 ATDs, its binding affinity to GluK1 ATD is significantly stronger than to GluK2 ATD. Similar results were obtained with GluK1 N72-137 and GluK2 N71-136. Moreover, in a control experiment, ATD of AMPA receptor subunit GluA2, did not show clear binding to GluK1 SP, suggesting the specificity of this assay. We also added the description and discussion of corresponding results in paragraphs on page 12.

Fig. 4 the interaction between GluK1 ATD and signal peptide. a.Co-IP. b. binding affinities.

Response to Reviewer #2's comments:

This is a nice paper following on from an important theme - unexpected complexity of

the molecular biology of glutamate receptor trafficking.

Response: Thanks for the nice comments.

The introduction is clearly written and introduces the topic well - however, I was surprised that the authors did not mention their own work on the AMPA receptor signal peptide until much later. I knew the work and therefore this paper made a lot of sense. However, a reader that did not know the existing finding that the SP can determine subunit dependent assembly in a related receptor could well be more resistant to this expansion of the canonical SP role.

Response: We thank the reviewer for this suggestion. We have added an introduction for “unconventional functions” of signal peptides on Page 4, and also referred our work about AMPA receptor signal peptide.

Overall the experiments are well designed and give considerable insight. However, the model put forward is quite challenging and therefore the authors should either provide more evidence, or scale back their conclusions and interpretation and be a bit more cautious, and identify more explicitly some of the (few) weak points in their interpretation.

Major points

I think the claim of a direct interaction between the cleaved SP and the ATD controlling trafficking, lacks for direct support. For example, the biochemical results in Figure 3F are really excellent, but they are on constructs that don't represent the real receptor and that might have rather strange properties. For example, the single TM domain of a receptor with multiple TM domains is not really guaranteed to be found in the membrane in the presumed manner. For these kind of “display” experiments, it would have been better to use a known single TM helix, like from Neto or pDisplay or similar. The authors could acknowledge this.

Response: This is a very thoughtful insight, thanks. We now realize that TM1 domain is not necessary for the Co-IP experiments and may add more complexity to the system. So we removed TM1 and repeated those experiments. The results are similar to that we previously showed (Please see Fig 4a shown above).

The key co-immuno precipitation experiment is different from the model the authors present. The experimental interaction here is of the non-cleaved SP (hindered with HA) attached to a membrane protein, with the ATD. That's rather different from the free SP (which has a different fate following processing) - and we don't know where this interaction (which compartment) is happening in this artificial system. Very difficult to assess, but for this reason, the conclusions must be more measured. The coIP doesn't, as the authors claim, show that “the GluK1 signal peptide directly interacts with the ATD and thereby represses GluK1 trafficking.” - only that the

hindered SP, when tethered to the membrane, can coIP with some part of the SPGluK2ATDGluK1-Flag-TM1 construct. Some parts of this construct may not be inert. This experiment would be much stronger if controlled against the K2 ATD.

Response: This is also a thoughtful insight. We now include GluK2 ATD as a control. We found HA-SP^{GluK1}-GFP could only weakly pull-down ATD^{GluK2}-FLAG. So we concluded that GluK1 signal peptide interacts more specifically to GluK1 ATD. We further test the direct interaction using fluorescence polarization experiments, and the results are consistent with Co-IP experiments, indicating GluK1 signal peptide directly interact with its ATD. Based on these updated results, we think it is much safer to propose “the GluK1 signal peptide directly interacts with the ATD and thereby represses GluK1 trafficking”.

The inhibition in trans (a very impressive result from Figure 1) brings an important point forward. To be certain that the effect of the trans-inhibition is on trafficking, it would be important to demonstrate that the surface expression was reduced in this experiment. It’s conceivable that the trans expression stops synaptic incorporation, without doing much to surface expression. This would be a further confounding element. Fusion of the GluK1 SP to mScarlet or a similar would allow imaging experiments to be redone with GFP tracer. In principle, the inhibitory SP could then be visualized at intracellular sites (again very challenging). Maybe just acknowledging this possibility is enough.

Response: There are actually two points raised by the review, both very important. The first point is whether the trans-inhibition of GluK1 signal peptide also works for surface expression in addition to its effects on synaptic trafficking. We therefore transfected cultured neurons with GluK1(SP^{GluK2}) (HA inserted after signal peptide for immuno-detection) or together with SP^{GluK1}-GFP and found that surface expression of this GluK1 chimera was reduced by co-expressed GluK1 signal peptide, indicating that the trans-inhibition is also involved in GluK1 surface expression (Fig. 5c and 5d).

Figure 5c. Surface expression of GluK1(SP^{GluK2}) is inhibited by SP^{GluK1}-GFP in trans.

The reviewer also raised another question as to the exact intracellular sites the SP-inhibition occurs in neurons. We appreciate the reviewer's nice consideration that this is very challenging especially for the technical limitation in imaging currently. We discussed the unknown mechanisms of GluK1 inhibition in revised MS (last paragraph in discussion).

There is not enough information about the sequences used. The authors say that they inserted the HA tag before the signal peptide (p6). It would be important to see the exact sequence. It seems hard to do this without disrupting the SP itself but perhaps sharing more of the sequences would make it clear.

Response: Thank the reviewer for this suggestion. Indeed, our previous experience indicated that tagging HA directly after signal peptides might interfere the expression or trafficking of kainate receptors (Sheng et al., 2015). Therefore, we applied small linker sequence "GGGS" in some of our constructs. Now the sequences of HA-GluK1, HA-GluK2, SP^{GluK1}-HA-GluK1, SP^{GluK2}-HA-GluK2, SP^{GluK1}-GFP, SP^{GluK2}-GFP, HA-SP^{GluK1}-GFP and HA-SP^{GluK1}-GFP in Fig. 2 are showed in supplementary Fig. S1. Moreover, in the revised Fig. 1 and Fig. 2, we included schematic cartoons for the chimeric constructs for clearance.

Minor

There are numerous grammar mistakes: e.g. "The GluK1 signal peptide and ATD are interacted directly" that could be fixed. A native speaker (for example, the prepenultimate coauthor; other authors may also be native speakers) could help with this.

Response: We thank the reviewer for this suggestion. We have corrected the grammar mistakes in current version.

Response to Reviewer #3's comments:

The manuscript by Duan and colleagues examines the role of the kainate receptor (KAR) signal peptide in regulating the targeting of GluK1 and GluK2 KA-type ionotropic glutamate receptors to synapses of CA1 hippocampal neurons. Previous work from the senior author has shown that GluK2 receptors readily target to the Schaffer collateral-CA1 synapse in a manner independent of the auxiliary Neto

proteins, Neto1 and Neto2. In contrast, GluK1 expression is much more limited and relies almost exclusively on Neto1 and Neto2 for trafficking and synaptic targeting (Sheng et al, 2015, Elife). A follow-up study then showed that this distinction between GluK1 and GluK2 is dependent on the amino-terminal region of each subunit ATD (Sheng et al, 2016, PNAS) though the exact mechanism by which the ATD region mediates this effect was not identified. In the present study, the authors attempt to dissect apart this mechanism(s) by looking more closely at the respective signal peptides of each subunit. The authors postulate a non-canonical role for the signal peptide of GluK1 which they propose directly interacts with the ATD region to inhibit the surface expression and synaptic targeting of the GluK1 receptor.

Although the study is of potential interest to neuroscientists interested in the biochemical make-up of central synapses, the manuscript suffers from several weaknesses that have tempered my overall enthusiasm for it. First, the authors use 3 different expression systems (organotypic cultures, primary cultured neurons and finally HEK 293 cells) displayed in 4 figures to study the role of the signal peptide. The underlying assumption is that the biochemical processes that regulate GluK1 trafficking/targeting are the same in each model system, however, there are many reasons why this might not be the case. Consequently, the authors need to document why switching between 3 very different model systems is valid - ideally by showing the same primary observations in all 3 systems.

Response: To study intracellular trafficking mechanisms, different systems are always applied as they all have experimental advantages as well as limitations. Synaptic response analyses of neuronal connection strength should, of course, be done in organotypic slice cultures as the neuronal circuits are well reserved. However, permeabilization efficiency of the slices is relatively low and it is much better to use primary cultured neurons for the analyses of subcellular localization of receptors. The HEK 293T cells are applied for biochemistry analyses as the transfection efficiency on culture slices or primary cultured neurons is too low to be practical for biochemistry analyses such as Co-IP. We should emphasize that in our study here, the results from the three cellular systems as well as the direct biochemical binding experiments are generally in consistent with each other.

Second, in terms of novelty, the authors' study is more like an extension of the previous published studies (in Elife, PNAS and JBC) looking at KAR trafficking. The unexpected role of the signal role of the signal peptide is novel, however, as discussed below, the authors need a stronger dataset to make this point.

Response: As the reviewer acknowledged, the non-canonical role of signal peptide revealed by this work is novel. We appreciate that the reviewer has pointed out this. We are confident that all our collected experimental data now have solidified this finding.

Third, the text of the manuscript is very unclear in many sections which made it

difficult to fully understand the authors' arguments/rationale. The manuscript would benefit from a careful review of the text to improve clarity for the reader. I have included other major concerns below to help the authors in the revision of their work.

Response: We thank the reviewer for this suggestion. We have polished the language and added some cartoons for illustration. We wish it would be more clear and helpful for the readers.

Major

1. Manuscript text. The text in general has a lot of awkward language and/or poor sentence construction. For example, the following sentence from the "Introduction" I think should read, "Both surface (EXPRESSION) and synaptic trafficking of (THE) GluK1 receptor fully relies on auxiliary Neto proteins, while GluK2 itself traffics to the surface and(/OR THE) synapse independent of Neto proteins ". I found similar problems in many areas of the Results section of the manuscript which made it difficult to read. For example, I am still not clear what the authors are referring to in this sentence from the Results section, "Therefore, we coexpressed the GluK1 signal peptide together with GluK1(SP2) in the same neuron and then examine the effect on synaptic transmission. To this end, we put GluK1(SPK2) after IRES to reduce its expression and found this GFP-IRES-GluK1(SPK2) construct still significantly increased synaptic responses in CA1 cells (Fig. 1C)". The authors need to revise the entire text of the manuscript thoroughly.

Response: We are sorry that there are mistakes in our manuscripts. We have corrected these mistakes and revised the text.

2. Model system. CA1 neurons express KARs (Castillo et al, 1997; Bureau et al, 1999) but not at the Schaffer collateral synapse which begs the question why they are not targeted to this synapse. The authors have shown that they can achieve this but it requires the overexpression of GluK1 (with Netos) and GluK2. Consequently, it could be concluded that the authors are studying an assembly mechanism that only occurs when the KAR proteins are expressed in abundance – which apparently isn't the case for wildtype CA1 neurons. Also, it seems unlikely that the mechanism responsible for endogenous KAR expression/targeting is due to the signal peptide so what might be going on? The authors need to address these confounding factors since they undermine the conclusions that can be drawn from the authors' work. Also, how do the authors know that GluK4 and GluK5 receptors are not contributing to the responses observed in the CA1 neurons? GluK4 and GluK5 only reach the cell's surface when co-expressed with GluK1,2 or 3. Consequently it is possible that the receptors may be heteromeric in nature – possibly containing GluK4/5.

Response: The reviewer raised two points here. First, the reviewer pointed out neurons express heteromeric receptors and our overexpression systems might not be natural. We fully agree with the reviewer that neurons express the heteromeric kainate

receptors containing GluK4 or GluK5. However, to experimentally define “natural receptors” is not easy. Because the native kainate receptors also contain auxiliary Neto subunits, other binding proteins such as C1q family proteins, and possibly other unknown factors. All above molecules could affect the receptor trafficking, and we don't believe it is possible to include all those “native” factors in one study.

As the reviewer acknowledged, we are using an over-expression system, where the endogenous kainate receptors should not play much as the endogenous kainate currents is undetectable in CA1 synapses. Therefore, on this “null-background”, we could study the assumingly pure population of homomeric receptors and focus on the interaction of GluK1 and its signal peptide. Our focus on this topic should not undermine the significance of our study, just like our work should not undermine others' work on the effects of Neto or of GluK4/GluK5 or of C1q family proteins on kainate receptor trafficking. We also discuss whether signal peptide might also affect heteromeric receptors in neuron.

The reviewer also raised a 2nd point; he/she worried about that the observed currents might be contaminated by heteromeric receptors which was assembled with endogenous GluK4/GluK5. Since there are no pharmacological tools that can separate those homomeric receptors and heteromeric receptors, we could not completely exclude this possibility. Based on our previous findings that the synaptic currents by kainate receptors in CA1 neurons are very fast (Sheng et al., 2015 and 2017), which is clearly different from slow heteromeric one in CA3 neurons (Sheng et al., 2017), we believe that the contamination should be very limited. To understand the GluK4 or GluK5 roles in trafficking might require overexpression of GluK4/GluK5 together with GluK1/GluK2, which is beyond the scope of our current study here.

3. Control cell recordings. The authors record from both transfected and non-transfected in the same organotypic slice preparations to compare their responsiveness. It would be expected that the responses from untransfected or control cells would be constant across all the conditions in the manuscript. However, a perusal of the scatter plots in figure 1 and 3 suggests that the response amplitude is quite variable. For example, a comparison of the data shown in Figure 1A suggests that the control KAR responses are less than about 100pA however in other panels (e.g. 3c and 3d), the responses can be up to 300 pA. The authors need to explain the variability in the control responses. One possibility would be to show (perhaps in a supplemental figure) a statistical comparison of the control responses across all the conditions. Given the nature of synaptic transmission, the response amplitude is probably not going to exhibit a normal distribution. As it stands, the variability in the control responses adds further to the uncertainty of the authors' findings.

Response: In this study, we used dual whole-cell recordings to analyze the synaptic responses by patching an experimental neuron and a neighboring control one simultaneously. Since these two neurons are very close, we assumed they had similar presynaptic input. Then any change of the synaptic responses should be attributed to

the treatment on the experimental neurons. This method has been used by Nicoll lab and other labs during last 15 years to study synaptic transmission as well as plasticity in hippocampus. The reviewer is right that the response amplitudes of control neurons vary a lot probably due the following reasons: 1) the detailed connection numbers vary among different slices although the general circuits are similar; 2) the distance of the stimulation electrode from the patched neurons vary among different experiments; 3) The stimulation strengths are not the same among experiments. However, the absolute amplitude values from different experiments are not critical for the final analyses and conclusion. The most important parameter is the relative responses or amplitude ratio between experimental and control neurons in each single experiment. Therefore, the effect of experimental treatment would be identified even the amplitude values of control responses are different among several experimental groups.

4. Figure 1. The text describing figure 1 was quite confusing especially in the section referring to the constructs GFP-IRES-GluK1(SPK2) and SP(GluK1)-GFP-IRES-GluK1(SPK2). I think I eventually understood the authors' rationale but it took me a lot of time to work it out. Clearer language would be helpful here and in other sections of the manuscript.

Response: We have added some cartoons for illustration in the revised Fig. 1 and Fig. 2. And we wish them were helpful for the readers to understand the experimental strategy.

5. Figure 2. The authors use HEK293 cells to argue that the signal peptide of GluK1 is not degraded as would be expected from conventional cell biology work. Although the data is consistent with this argument, it is still does not address whether this occurs in CA1 neurons of the hippocampus. Since this point is critical to the overall argument developed by the authors, proof that the signal peptide of GluK1 is "spared" in native cells is needed.

Response: This is similar point as reviewer 2 raised in his 3rd major comments. Both reviewers wish to see the signal peptide in cells. However, this is impractical currently because of technical limitation and the reviewer 2 has also acknowledged "very challenging".

6. Intracellular staining. The authors argue that data in figure 4 demonstrates that different constructs differentially distribute between the intracellular compartments and plasma membrane. Although the staining pattern is clearly different between some of the constructs, I think it is very difficult to conclude that the staining which appears to be mainly intracellular is not on the cell's surface. Likewise, the appearance of surface expression can only be demonstrated conclusively by TIRF microscopy and, ideally, using a pH sensitive GFP-tagged receptor to establish that the KAR is on the surface of the plasma membrane.

Response: The immunostaining experiments were carried out with a two-step protocol. Impermeable staining was first applied to examine the surface expression level of each receptor. Then the same samples were permeabilized with TritonX-100 and the expression of the intracellular receptors was examined using a different antibody. The detailed protocol could be found in the section of methods. This method has been widely and traditionally used to examine the expression level of glutamate receptors in neurons, which could be referred to our previous study as well as many other labs.

Minor Comments

1. Introduction. The points raised in the first few sentences of the text need to be referenced.

Response: We thanks the reviewer for suggestion. The references have been added.

Reviewers' Comments:

Reviewer #1:

None

Reviewer #2:

Remarks to the Author:

First of all I would like to apologise to the authors for the delay in returning my opinion. The complex and novel content of the manuscript demanded a closer inspection than I had time for, until now.

I find the authors response compelling. I am seriously impressed by the extent of new and revised experimentation that is included. The authors ideas from the first version of the manuscript have stood up to strong scrutiny and are therefore very solid.

This is top quality work and I commend the authors for their commitment to the project.

I have one major comment and some minor comments on the much improved text.

Major comment:

The GST-tagged ATDs produced in E.coli (pages 12 and 20) are unfortunately not folded, even if electrophoresis says that they are pure. This is known since nearly 15 years and was the reason that crystal structures of the ATDs were only obtained when ATD constructs were expressed in insect cells or HEK cells (Gouaux, Aricescu, Greger and Mayer labs). These experiments with fluorescence polarisation therefore rely on interactions between peptides and, at best, partially folded ATDs. The unfolded character must be the case for the presumed "critical sub region", residues 72 to 137. These experiments still have value. But the authors must clearly state the limitation that the overexpressed proteins were likely not fully folded.

Minor comments

line 128 "should lead GFP synthesised in secretary pathway" - several mistakes here

line 156 "constructions" - constructs

line 181 "The rational is" - the rationale being

line 187 "Besides," - in addition

line 191 "GFP was only seen in the cytoplasm and was absent from the nucleus (Fig. 2f, b-e & b'-e'), indicating it is synthesized and secreted through the ER membrane system."

Is it cytoplasmic or in the ER lumen? This rather looks ER to me, which anyway would make more sense? Perhaps I didn't understand this logic.

line 340. To determine the above underlying molecular and cellular mechanisms would significantly advance our understanding on glutamate receptor trafficking in future.

— it's more common to say, "more work will be needed to determine the underlying mechanisms" than to proclaim how wonderful it will be when the work is complete.

REVIEWERS' COMMENTS:

Reviewer #2 (Remarks to the Author):

First of all I would like to apologise to the authors for the delay in returning my opinion. The complex and novel content of the manuscript demanded a closer inspection than I had time for, until now.

I find the authors response compelling. I am seriously impressed by the extent of new and revised experimentation that is included. The authors ideas from the first version of the manuscript have stood up to strong scrutiny and are therefore very solid.

This is top quality work and I commend the authors for their commitment to the project.

I have one major comment and some minor comments on the much improved text.

Major comment:

The GST-tagged ATDs produced in E.coli (pages 12 and 20) are unfortunately not folded, even if electrophoresis says that they are pure. This is known since nearly 15 years and was the reason that crystal structures of the ATDs were only obtained when ATD constructs were expressed in insect cells or HEK cells (Gouaux, Aricescu, Greger and Mayer labs). These experiments with fluorescence polarisation therefore rely on interactions between peptides and, at best, partially folded ATDs. The unfolded character must be the case for the presumed “critical sub region”, residues 72 to 137. These experiments still have value. But the authors must clearly state the limitation that the overexpressed proteins were likely not fully folded.

Response: Thank Andrew for pointing out this. We now discussed this possibility (Page 13, highlighted).

Minor comments

line 128 “should lead GFP synthesised in secretary pathway” - several mistakes here

Response: Thanks. Now corrected. Page 7 highlighted.

line 156 “constructions” – constructs

Response: Corrected.

line 181 “The rational is” - the rationale being

Response: Corrected.

line 187 “Besides,” - in addition

Response: Corrected.

□

line 191 “GFP was only seen in the cytoplasm and was absent from the nucleus (Fig. 2f, b-e & b'-e'), indicating it is synthesized and secreted through the ER membrane system.”

Is it cytoplasmic or in the ER lumen? This rather looks ER to me, which anyway would make more sense? Perhaps I didn't understand this logic.

Response: Thanks for pointing out this. GFP without signal peptides is synthesized in cytosol and freely move into nucleus. GFP with signal peptides should be synthesized in ER and is prohibited to enter nucleus. GFP with GluK1 or GluK2 signal peptide was kept out the nucleus indicating the GluK1/K2 signal peptide directed GFP into ER lumen. We now clarified this. (Page 10 highlighted).

line 340. To determine the above underlying molecular and cellular mechanisms would significantly advance our understanding on glutamate receptor trafficking in future.

— it's more common to say, “more work will be needed to determine the underlying mechanisms” than to proclaim how wonderful it will be when the work is complete.

Response: Thanks. We modified the sentence accordingly.